# Making Sense or Non-Sense? Communicating COVID-19 Guidelines to Young Adults at Danish Folk High Schools

**DOI:** 10.3390/ijerph20032557

**Published:** 2023-01-31

**Authors:** Thilde Vildekilde, Julie Jakobsen Connelly, Charlotte Sophie van Houten, Jakob Thestrup Hansen, Jane Brandt Sørensen, Dan Wolf Meyrowitsch, Flemming Konradsen

**Affiliations:** Global Health Section, Department of Public Health, University of Copenhagen, Øster Farimagsgade 5, 1353 Copenhagen K, Denmark

**Keywords:** COVID-19, qualitative research, risk communication, boarding schools, young adults, sense of community

## Abstract

Little is known about young people’s behaviors and responses under outbreaks of infectious diseases such as the COVID-19 pandemic, especially in institutional settings. This research investigated the reactions of young adults residing at Danish folk high schools (FHSs) towards COVID-19 guidelines and the communicative styles used to enforce COVID-19 guidelines. The qualitative data consists of focus group discussions (FGDs) with students, interviews with staff, and participant observations, as well as survey data from 1800 students. This study showed that young adults reacted negatively when first faced with the new reality of COVID-19 restrictions. They expressed distress over the loss of meaning (non-sense), loss of sense of community, as well as uncertainty. Hygiene guidelines, however, made immediate sense and were socially well accepted. Most FHSs actively involved students in risk communication and creative examples of community-building communication were identified. This study demonstrates that successful risk communication at educational institutions must take into consideration how young adults make sense of and cope with the uncertainties of life during crisis situations including epidemics.

## 1. Introduction

Educational and institutional settings such as universities and boarding schools have the potential to become settings for super spreading events during infectious disease outbreaks such as with the SARS-CoV-2 virus. This is due to high levels of close physical interaction among a large, social and mobile group of young individuals, as well as the fact that high rates of young individuals who were infected with the SARS-CoV-2 virus have turned out to be asymptomatic carriers. This enables the significant spread of infection before school management or authorities can react. Several studies from educational settings during previous viral outbreaks have shown that close-contact educational environments present exceptional challenges for disease control for school management [1]. This relates to ensuring isolation and distancing [2,3] along with consistent hygiene practices, such as hand washing [4,5].

In Denmark, the adult population in the age group of 20–29 years have had the highest rates of infection with the SARS-CoV-2 virus over the entire course of the epidemic [6]. The young adult population in Denmark and elsewhere has also received much media attention and been repeatedly publicly shamed for their behaviors when questioning or challenging COVID-19 rules and guidelines [7]. Despite this, little is known about young Danish people’s behaviors and responses to authorities’ control measures during crises, including the COVID-19 pandemic.

### 1.1. The Context: Folk High Schools

The folk high school (FHS) is a type of so called ‘non-formal adult education’, mostly found in the Nordic countries. With a non-academic approach, students are taught a wide range of subjects such as music, dance, outdoor living and sports, philosophy, journalism etc. A stay at a FHS is mostly self-paid and chosen by young adults as part of their self-development. The majority of FHS students are young adults aged between 18 and 24 years old. Most FHSs are boarding schools with large campus areas, where students generally live close together in shared rooms and have shared kitchenettes and common rooms. They eat all meals together in dining halls, are part of daily cleaning and cooking routines, and spend most of their free time socializing on the school premises. During the 2020 spring semester, approximately 5700 students were attending courses in the 70 FHSs present in Denmark [8].

Little contemporary empirical research exists from the FHS context, even though it is a popular school form for young adults across Scandinavia [9]. In particular, the FHS context offers refreshing perspectives on contemporary adult ‘life education’ and pedagogical innovation [10]. FHSs thus provide an excellent natural ‘laboratory’ for communicators to experiment with and learn about different pedagogical and communication styles in a highly socially dense environment with special challenges—including risks of infectious disease outbreaks.

### 1.2. COVID-19 Guidelines on FHSs

On the 27th of May 2020, Danish FHSs were allowed to re-open after lockdown, which meant that the spring batch of students could come back to the schools and continue their FHS stay, but under a set of ‘COVID-19 guidelines’ issued by the Ministry of Culture [11]. The main components of the COVID-19 guidelines included intensified hand-hygiene and cleaning routines for the entire school area (which students would take part in), and outbreak and containment strategies including dividing students into ‘primary groups’ or ‘family groups’ of a maximum of eight people, where close physical contact was permitted, and all meals and teaching took place. Physical distancing guidelines required students to maintain a minimum of a 1-metre distance between themselves and students from other primary groups. Similarly, teachers were required to maintain a 1-metre distance from all students and to eat separately. All meals, teaching, and social activities were re-organized according to these guidelines, which meant major changes to all aspects of daily life at a FHS. All FHSs were also encouraged by the authorities to recommend staff and students to be tested for SARS-CoV-2 virus and to implement intensive COVID-19 communication strategies to raise awareness of disease control measures among all students, including introductions to the new COVID-19 guidelines issued by the authorities.

### 1.3. Research Aims

In this study we investigated the COVID-19-related perceptions of students at FHSs and explored the communication and pedagogical styles used at FHSs to enforce the COVID-19 guidelines. Data were collected during the first five weeks after the re-opening of the FHSs in May 2020. This allowed us to explore early reactions and how young Danish adults make sense of infectious control measures and restrictions communicated by the authorities and management within an educational setting. These insights can provide new perspectives on how to design future communication strategies to promote the control of a wide range of social and health crisis situations, including infectious disease outbreaks, in boarding schools and other institutional settings, including how to make sense and address resistance, uncertainty and frustrations among young people.

### 1.4. Theory: Sense-Making in Times of Crisis

During times of disaster, crisis, and major societal disruption people often experience a loss of predictability, certainty, and coherence in life. As Castiglioni and Gaj [12] describe in their study on sense-making of the COVID-19 pandemic, the COVID-19 crisis is a major disruptive event and thus a signifier for the ‘unknown’ and ‘the uncertain’, defying all our usual ‘sense-making’ and demanding reconfiguration and adaptation to new meanings. The study is inspired by the construct of sense of coherence as first proposed by Antonovsky [13] and since applied to cases of emergencies [14,15]. Castiglioni and Gaj see that the ability of individuals and groups to deal with the direct and indirect after-effects of a pandemic such as COVID-19 needs to take into consideration how we as humans form coherence and meaning out of seemingly ‘senseless’ situations. What might be perceived as ‘not making sense’ in the early phases of an emergency, needs to be reconfigured to make sense of being able to operate in a ‘new normal’ scenario. Castiglioni and Gaj mention that a process of ‘normalization’ of the expected emotional reactions of uncertainty, disappointment, anger, nervousness, etc. is vital to address in response strategies. This includes the usefulness of making new routines and the construction of consistent metaphors and narratives in order to explain what is going on, as these can help reinstall predictability and re-create a sense of coherence, meaning, and group identity in affected population groups [12]. In addition, Castiglioni and Gaj [12] underline the potential of unclear or conflicting messaging from authorities to challenge feelings of certainty, predictability, and manageability, which adds to feelings of ‘non-sense’ among target groups.

Most young adults in Denmark have not faced a societal disruptive event similar to the COVID-19 pandemic and have not previously been forced to change and adapt at this scale. For the first time since the 1950s, a pandemic forced all Danish educational institutions, including FHSs, to impose a set of infection-control and preventive restrictions on students’ lives. During the analytical process in this study, we found two core empirical themes emerging around: (1) ‘Restrictions and messages being perceived by students as non-sense’ in the initial phase of adjustments’; and (2) ‘Restrictions as a practice destroying the FHS community’. We find that these themes can be meaningfully explored by applying the above-mentioned theoretical concepts of ‘sense-making’ and ‘sense of coherence and community’.

## 2. Materials and Methods

The overall paradigm guiding all data collection and analysis in this research project was a pragmatic one, which applies a flexible approach to the research design and analysis [16]. Following the pragmatic approach, a mixed-methods design was chosen, allowing for the opportunity to investigate the research questions with multiple methods, both quantitative and qualitative. Specifically, we focused on a particular social domain (the FHS) and ‘the social problem’ of disease transmission in this domain [16]. This paper only draws on data of a qualitative nature from several data collection methods as described below.

### 2.1. Qualitative Data from a Student Survey

A total of 16 FHSs accepted to be part of this study. A series of online surveys available in Danish and English were administered to staff and students in all 16 FHSs 3 weeks into the re-opening (from 3 June until the 9 July 2020). We received a total of 1834 answers from students and 679 replies from staff. In this article, only data from two open ended survey questions posed to students will be used, as they relate directly to the research focus and can detail and verify the findings from the other qualitative data collection methods. The two survey questions were: ‘What could motivate you to follow the COVID-19 guidelines even better?’ (357 responses received from students); and: ‘Describe your experiences and/or opinions regarding the measures that have been put in place at the folk high school in order to prevent spread of the coronavirus’ (375 responses received from students).

### 2.2. Qualitative Data Collection at FHSs

Out of the 16 included FHSs 5 were selected for further qualitative data collection. The selection was based on a combination of availability on specific dates, diversity in the types of schools (sports, arts, and other types), location (schools in the western, central and eastern parts of Denmark), and the type of student population (Danish as well as international students). A minimum of 5 h was spent at each school by a team of 1–4 researchers during a weekday. Three schools could accommodate the team for longer or several visits lasting between 16–24 h over two different days including weekends. This allowed the team to observe a wider variety of COVID-19-related behaviors. At each school an observational tour was conducted, guided by a staff member, investigating the new COVID-19 living, eating, teaching and social activity arrangements, handwashing and cleaning facilities, routines, and studying visual and informational materials to guide behavior including distancing. The research team joined meals and activities during the visits and conducted participant observations focusing on social behavior and physical distancing.

A total of 9 mixed-gender focus group discussions (FGDs) were conducted with a total of 38 students (22 female, 16 male), the majority of them being in their early twenties (age span from 18 to 32 years), mainly including Danish students (*n* = 23), but also students from other Scandinavian and European countries (*n* = 7) and Asian and South American countries (*n* = 8). The participant sampling was based on availability and snowballing for more participants. Six FGDs were conducted as formal sessions with audio recordings, while three sessions were informal and took place during a leisure activity at the school with no audio recordings. All FGDs followed a thematic guide with three overall themes including (1) perceptions of the COVID-19 disease and its risks, (2) attitudes towards the COVID-19 guidelines, and (3) experiences of the new life at FHS with COVID-19 guidelines, particularly related to hygiene practices, social and emotional life, and social distancing.

One semi-structured interview was conducted with the school principal or another leading member of management at each school to understand (1) the communication approach taken and the materials used at the school to guide behaviors, and (2) the processes and challenges of implementing the COVID-19 guidelines at the school. Interviews and conversations were conducted with an additional 13 staff members across schools.

Verbal consent was obtained from all the interview participants and FGDs prior to data collection. The participants were informed about the purpose of the research and that participation was voluntary and anonymous. For the sake of anonymity, the name and location of the schools, as well as the nationality of the informants will not be mentioned in this paper.

### 2.3. Data Analysis

Compatible with the pragmatic paradigm, the analytical process followed the principles of an integrated or abductive approach, moving forth and back between inductive and deductive reasoning [16,17]. Thus, the team performed a first round of analysis after each FHS visit, identifying and describing the broad empirical themes directed by the interview guide. In a second round of analysis, a thorough reading of all observation notes, FGD and interview transcripts, and open survey answers was conducted and the focus intensified on central analytical and theoretical concepts such as ‘sense-making of guidelines’, ‘distancing’, ‘loss of community’, and ‘perceptions of hygiene rules’. The interview and FGD transcripts were coded by one author and the initial themes and analytical possibilities were then discussed with the rest of the team by presenting them with extensive tables with quotes, theoretical papers, and concepts that could facilitate the analysis.

The open-ended survey answers (which were mainly statements of 1–3 sentences) were coded and assigned into themes by two authors in parallel and then reviewed and discussed by the two authors to ensure consistency. Codes were firstly organized under broad descriptive themes following the interview guide but also creating new themes as they emerged during the coding process.

## 3. Results

### 3.1. Students’ Reactions to COVID-19 Guidelines: Making Sense or Non-Sense?

The majority of the interviewed FHS students expressed an understanding of the importance of preventing transmission of the SARS-COV-2 virus to the wider society by following certain behaviors and rules at the FHSs, as they agreed that COVID-19 was a serious disease, particularly for older and vulnerable population groups. However, the majority of the interviewed students also expressed high levels of frustration with the COVID-19 guidelines at the FHSs, perceiving them as ‘meaningless’, or ‘not making sense’. As Table 1 shows, 90 out of 357 survey answers (25.2%) stated that if the rules ‘made more sense’, or fitted more logically into the FHS context, it would be more motivating.

Another aspect of meaninglessness, mentioned in the survey as well as FGDs, was the perceived low credibility of the guidelines due to mistrust of the authorities issuing the guidelines—including political actors and health authorities. It was repeatedly mentioned that ‘authorities don’t know anything about FHSs’, hence the guidelines were perceived as a disruption to ‘the normal FHS’ way of life. The fact that special, and to some extent more strict guidelines were issued for FHSs compared to ‘outside society’ also undermined the credibility of the guidelines, and the fact that the guidelines were interpreted and implemented differently across different FHSs created frustration. For example, at the time of re-opening, the rest of society followed a 15-person maximum recommendation for social gatherings, while FHS students were only allowed to interact closely with eight people. In addition, some FHSs had guidelines for students to minimize contact with the surrounding community, while the wider population was still free to attend fitness centers, go shopping etc. Guidelines about ‘partying’ were also vaguely formulated which gave the FHSs room for different interpretations and locally adjusted solutions. Some illustrative quotations are shown in Table 2. Overall, the necessity of adaptating behavior to crisis by following new rules was met with substantial emotional frustration and feelings of meaninglessness, uncertainty, and a demand for more clarity and ‘sense-making’ by FHS students.

### 3.2. A Democratic Dilemma and A New Sense of Community

In particular, the issue of reaching consensus on solidaric behaviors and the consequences of not following guidelines created dilemmas among students (see Table 2). For example, students at all FHSs were free to leave the school premises daily, which led to discussions in some primary groups about the risk of exposure to infection of the FHS community and of the FHS closing down again. Many students therefore requested a higher degree of stringency and consistency of guidelines to prevent conflicts and to counter their own sense of uncertainty, while only a few students requested a hard line that allowed for direct consequences and punishments for not following the guidelines (see Table 1). At the same time, many students in FGDs and the survey expressed that they wanted a high degree of involvement of students and the freedom to be critical towards norms and COVID-19 rules, which was seen as being at the core of the ‘old’ FHS community (see Table 2). A small number of the interviewed students said that the guidelines and restrictions patronized them and made them feel they were ‘being controlled like children’, and many students reflected on the difficult new role of FHS teachers as ‘policing’ student behaviors (See Table 2). Hence, the COVID-19 crisis has highlighted a real democratic dilemma for FHSs: the FHS is an institution that aims at teaching young adults life skills, including democratic values and mindset, encouraging diversity in views and behavior. However, because of the outbreak, a certain degree of alignment and enforcement of behavior was needed for effective disease prevention and control. This gave students a feeling of facing a ‘new normal’ sense of community guided by the values of social responsibility, democracy, and trust.

### 3.3. Physical Distancing Challenging the Sense of Community

In line with the wishes of most FHS students to assist in effectively preventing the spread of COVID-19, our observations at FHSs showed that physical distancing guidelines were mainly complied with during formal activities such as mealtimes and class-based teaching activities. Distancing guidelines were hard to follow during other types of courses, including playing in bands, performing theater, singing in choirs, and doing gymnastics. Our observations during evenings and weekends also showed that distancing rules were typically not followed. We, amongst others, witnessed informal pool parties, groups sitting and laying on couches closely together, large groups swimming and dancing together, and small gatherings developing into all-night parties with lots of physical interaction. We also heard students talk about ‘secret parties’ and romantic partners who broke the rules to spend time together. These observations were supported by the interviewed students and staff.

Physical distancing was described as the most difficult practical aspect of the new COVID-19 guidelines. The division of all students into eight-person primary groups to minimize physical contact was therefore the most disliked part of the guidelines by both students and staff. Most students in the FGDs and a total of 120 (33.6%) (see Table 1) answers from the survey expressed negative opinions about these primary groups. Distancing was often described as ‘impossible’, ‘hard’, ‘unrealistic’, or ‘unfeasible’ to comply with in the FHS context they knew pre COVID-19. The distancing guidelines were therefore also often viewed as ‘meaningless’, ‘nonsense’, or ‘pointless’ in relation to the known FHS life and sense of community. Overall, the students expressed that the consequences of distancing included a sense of separation and a loss of community feeling, which the FHSs were traditionally fostering, and which students had experienced during their pre-COVID stay. In all the FGDs and in 25 survey answers, students said that physical distancing was destructive for the ‘folk high school spirit’ and its feeling of togetherness. Especially in the survey, the students expressed strong emotional distress about this sense of separation from others, resulting in feelings of stress, anger, hatred, sadness, anxiety, demotivation, sorrow, loneliness, and hopelessness. Table 3 shows examples of the many quotes about physical distancing. As shown in Table 1, and backed up by students in all FGDs, most students therefore wished that all students at a FHS could be allowed to function as a ‘closed social bubble’, or at least with larger primary groups, and with no or little contact with the outside community, allowing for more closeness and community feeling in the FHS.

### 3.4. Hygiene Guidelines Make Sense

As opposed to the strong resentment and many negative emotional reactions towards the rules on distancing, the students’ attitudes towards the increased focus on and their involvement in improving hygiene standards at FHSs were positive and intuitively made sense to most. The students expressed that the increased focus on hygiene ‘made good sense’ and it was widely accepted and appreciated, and quickly became ‘the new normal’ in daily life. The students also expressed that being ‘forced’ to learn how to clean more often and rigorously in common facilities such as toilets and baths, dining areas, common rooms, and teaching facilities provided a new and more comfortable living environment, though the daily cleaning of private rooms was seen by many as ‘excessive’ and ‘annoying’. All the students in FGDs also stated that due to the close living arrangements and a lot of physical contact, infections used to spread quickly and thrive in the student population, and they speculated that improved hygiene would also have a positive effect on preventing a lot of common illnesses and sick days. Interestingly, the international students interviewed at two of the FHSs felt that the hygiene standards at these Danish FHSs needed this lift, especially in terms of improved hand hygiene. Illustrative quotes from students are shown in Table 4.

### 3.5. Strategies to Communicate COVID-19 Guidelines: Individual Responsibility, Democracy, or Control?

The core of Scandinavian FHSs is the democratic nature of adult education, characterized by a high degree of personal freedom and democratic decision making. Thus, the new guidelines were received with some concern at FHSs, since a rule-based culture seems to go against the very heart of the ‘Folk High School spirit’ and learning style, while also potentially disrupting the equal relationship between students and staff. It was therefore in line with our expectations to find that all five principals greatly emphasized the role of student responsibility, empowerment, and dignity, while also communicating the necessity of agreeing on the new guidelines to students. One principal said: 

*“We constantly have to remind each other: Why are we here? What are we doing? How do we communicate with the students? How do the students feel empowered through a stay here instead of the opposite, just being told what to do or not? […] They (the guidelines) really sort of raised the question, ‘what is a Folk High School?’. How do we ensure that we stay in touch with our students, and that they still respect us?”* (Principal, FHS2). 

The principal in FHS 1 similarly stressed the need for fostering dignity in students when introducing the guidelines: *“This conversation about how we do this in a way that makes students feel respected and dignified in this process. Because that’s in our DNA—it’s our values as a folk high school”*. Other staff and principals stressed the importance of teaching primary groups to take responsibility for each other and take responsible decisions, treating students as adults, and appealing to students’ sense of responsibility and community. In this sense, the pandemic forced the FHS community to reflect on their identity as FHSs—which type of community they would like to represent, and which role to play in society during times of changes. We found that in all five FHSs the management and teachers had incorporated elements of interaction and student involvement in various forms when communicating the guidelines. This included ad hoc and planned meetings with students in large and smaller groups, debates about experienced dilemmas and conflicts during daily life at the FHS, facilitating consensus seeking processes about social behavior, including parties and drinking, and regular feedback to students on new initiatives or changes to the guidelines. Two FHSs took a more directive approach to introducing and enforcing the guidelines, that to a smaller degree allowed the students to debate or change the guidelines. Three FHSs took a democratic approach, accommodating the students’ views and wishes about the guidelines and ‘encouraging’ adherence.

### 3.6. Creativity and Behavior Design

All the visited FHSs were ambitious about fulfilling the COVID-19 guidelines, but also challenged in terms of establishing and inventing effective COVID-19 risk communication and pedagogical content in only a matter of days. Four out of five FHSs had applied creative pedagogical tools to communicate and discuss the guidelines, including role plays, songwriting, making poetry, and designing COVID-19 memes, and posters. However, only two FHSs used this opportunity to integrate behavior design strategically to nudge hygiene and distancing behavior to prevent infection and outbreaks. They had designed pictograms, jingles, and stickers, and used effective color schemes and positive messaging about maintaining distance, washing hands, disinfecting ‘hotspots’, etc. The experiences with creativity and behavior design were positive at the FHSs. It took the pressure off staff from ‘reminding’ the students and instead introduced humor, conversation, and interaction. It also gave the students the opportunity to express their frustrations and questions through various types of creative outlets and not only through confrontations with staff.

### 3.7. Narratives and Metaphors to Support ‘Sense-Making’

As stated in the introduction, narratives and metaphors can be used strategically to re-install or create a new sense of meaning and community, when ‘normal life’ has broken down. Two principals presented the guidelines to the students as ‘made by government—not by us’ and as ‘non-negotiable’. This gave the students and staff little possibility to interact, discuss, identify, and make new meanings out of the situation and the guidelines. A different approach was taken by two principals, who worked consciously to create and use positive and community-based narratives and metaphors to create a new sense of coherence for the students. The management at FHS1 worked according to the metaphor of ‘being pioneers’, showing the way for future FHS students: 

*“I talked to them about that they should see themselves as pioneers in this new way to be on a FHS. They will be laying the stepping stones down for, not only this semester, but also the following semester […] So the solutions they come up with together with us, will have much broader consequences […] they are the ones who need to find the solutions together with us”* (Principal, FHS1).

The principal at FHS3 created a narrative about ‘Heaven and Hell’ inspired by the French author Albert Camus: 

*“I used it as a story or a narrative about leaving the school area. I encouraged them to think that ‘it’s the people over there in the shopping center who are in Hell, and this place is Heaven’. It was a way to focus on here and now, and the more of a social bubble we make here, the more protected we are from infection. And they want that—they don’t need the outside world at all!”* (Principal FHS3). 

Both narratives centered around creating a sense of community, and ‘being in it together’, creating a new sense of group belonging. Three principals had deliberately chosen not to draw on the narrative broadly used by the Danish Government at this point in time, which was to encourage all citizens to demonstrate a ‘societal mind’ or ‘samfundssind’ in Danish, complying with the guidelines to protect the elderly and vulnerable in society. Although we found that the students in general understood this argument as valid for preventing disease outbreaks, the narrative was not seen by the principals as effective reasoning to change young adults’ behavior: 

*“In the beginning I was much more into this story of taking responsibility for the elderly and wanted to build on that story […] But they have heard it too much. We have to build the story in another way that is much closer to where they are in their lives and this phase of life they are in. And I think that was a very clever decision”* (Principal, FHS1). 

These experiences show that developing effective communicative narratives can assist in building a new sense of community and belonging and can be a starting point for a neo-normalization and acceptance of ‘the new way of life’ in the FHS context. This is important for future epidemic communication strategies.

### 3.8. COVID-19 as an Opportunity for Learning

The management representatives at all the visited FHS were asked how and if the COVID-19 crisis could be an opportunity for learning, e.g., leveraging discussions with students or changing old habits. Only two examples of the COVID-19 pandemic being used directly in teaching were mentioned across the five FHSs (i.e., discussing the global politics and media coverage of the pandemic). The staff felt that the students were ‘corona-fatigued’ and did not want to engage with content about the pandemic. However, several of the interviewed teachers and staff indicated that the guidelines had forced them to think outside of the box and create new teaching styles and solutions, including more mentoring and use of online tools, asking the students to assist each other and act as co-teachers, as well as new meal and cooking routines. The principals at all five FHSs agreed that an improved focus on hygiene was a good result of the crisis for general health and wellbeing at the FHS, and initiatives such as hand sanitizers and hand washing basins were seen as useful also after the COVID-19 crisis. At FHS 5, which had a large international student population, the COVID-19 pandemic created a new opportunity to discuss cultural differences in the perceptions of acceptable social behavior, including distancing and bodily boundaries, partying, and drinking alcohol, thus strengthening cross-cultural awareness in students. Most clearly, the interviewed staff at all 5 FHSs highlighted that the COVID-19 crisis was an (inconvenient) opportunity to teach life skills and enable the maturing of students, by taking responsibility, showing solidarity, realizing the wider consequences of one’s own actions, and learning to cope with the uncertainty of life and restrictions on individual freedom and choice. As one staff member stated: 

*“We should see it (the corona crisis) as a challenge for ourselves and that we can use it to get creative, overcome challenges, and find solutions. This actually talks very much into the ‘folk high school idea’—it’s about taking responsibility—not just for ‘myself’ but for others. And about knowing the wider consequences of my own actions […] This generation thinks that everything is up for discussion and can be debated. But now there are just strict rules and that is a challenge to them!”* (Staff, FHS 3). 

The principal of FHS 4 also said; “*Many students thinks that this (living with restrictions) is hard. And then we just agree with them that ‘yes its hard’ […] This is about maturing for these young adults aged 18–22. It’s about learning that things in life can be hard, and to accept that it’s hard right now*”.

## 4. Discussion

Since this research and its outcomes were of high and acute relevance to the FHSs in their ongoing COVID-19 response during the summer and autumn of 2020, the preliminary findings were communicated to the FHSs through several written popular media channels. The research team also attended two online meetings with principals from the majority of Danish FHSs, a large group of pedagogical staff, and technical staff, in which the findings, recommendations for action, and implementation challenges were discussed.

Based on the findings of this study, we wish to discuss four main issues.


**Issue 1: Should hygiene be put back on the agenda for educational institutions?**


Yes! As mentioned in the introduction, we know that boarding schools in different cultures have experienced special challenges during epidemic outbreaks—especially in terms of maintaining appropriate hygiene. Hygiene interventions have shown to be effective in reducing illness in educational settings with children [18,19]. In Denmark, boarding schools for secondary students and FHSs are settings where students live and study closely together, which may facilitate easy disease transmission. However, hygiene has until now been an under-prioritized area in Danish educational institutional settings. This research showed that hygiene promotion is a low-hanging fruit in disease prevention; improving hygiene standards at FHSs intuitively made sense for students, was widely socially accepted by all, and was easy to implement and integrate into ‘a new normal school life’. We therefore recommend that boarding schools and educational authorities put hygiene back on the agenda—not just during times of epidemics, but during normal school life, potentially preventing a large range of infectious diseases as well as increasing feelings of wellbeing. Smart behavior design can be used to effectively nudge hygiene behavior [20].


**Issue 2: How can we understand young people’s reactions towards COVID-19 guidelines?**


Denmark has not been affected by any comparable large scale epidemic outbreaks such as COVID-19 or any major societal disruptive events since the Second World War. Danish society is today characterized by being an individualistic, high-income setting, exhibiting a high degree of democracy [21] and stability [22], and low levels of vulnerability [22], corruption [23], and inequality [24]. Furthermore, Denmark can be described as a western individualistic and ‘loose’ culture with high degrees of personal freedom, non-strict social norms and low social control [25]. Many young Danish adults may therefore not be used to coping with and adapting to sudden disruptive life events, or to accepting restrictions on their personal freedom. The perceived loss of meaning, control, and sense of coherence that we identified among young adults in this study during the first weeks of the COVID-19 pandemic in Denmark may therefore be interpreted as the result of living in a stable, secure, individualistic, and free cultural context. These young people are thus encountering un-expected, un-controllable uncertainties, and the authoritative responses to an epidemic for the first time. This included accepting authorities who normally do not interfere with individual and social behavior (such as party and drinking behavior) and accepting the changing nature of rules that were designed and implemented unexpectedly and rapidly. This was despite the fact that the Danish authorities as well as school management, took a ‘loose’ approach to governing behaviors during the coronavirus crisis by issuing ‘guidelines’ rather than strict laws, which tends to be normal policy practice in ‘loose’ cultures [25].

This cultural practice was confirmed by the data from international students coming from collectivistic and more ‘strict’ authoritarian cultures with previous experiences of epidemic outbreaks. They expressed confusion about the ‘loose’ approach and a high acceptance towards strict social and behavioral norms.

This paper focused on a specific school context of FHS in a Danish cultural context, which may mostly be comparable to an equally free school context such as American colleges. However, the findings demonstrate how important it is for any educational facility, and socially dense boarding school contexts in particular, to be aware of and manage the divides between individualistic and collective norms and the interests of young people during emergencies. Being in a highly collectivistic context (a boarding school setting) embedded in an individualistic culture, creates a special democratic paradox that boarding school institutions across western cultures may face.

Independently of cultural context, however, it is a challenge for future crisis management and disease control among young adult populations, if guidelines are generally perceived as ‘meaningless’ or ‘not making sense’ by them. Therefore, a process of ‘sense-making’ could beneficially take place at all educational institutions during any kind of future emergencies, including outbreaks of infectious diseases. This must include students and staff together to create new meanings in the new reality that they face. As mentioned by Castiglione and Gaj [12] and as demonstrated in this study, schools such as FHSs could benefit from engaging in such a normalizing process, drawing on new meaning-making and community-building narratives or metaphors. In this way, we see that Danish FHSs and other educational institutions can play an active role in assisting young adults to develop new coping strategies when facing the uncertainties of life during virus outbreaks and other types of emergencies.


**Issues 3: How do we manage social distancing in hyper-social educational contexts?**


Young adulthood is a period in life in many different cultural settings characterized by a lot of social interaction. The FHS context in Denmark, and other types of boarding schools in other cultures, is a hyper-social context where this is experienced to the fullest. This study showed that the loss of contact linked with physical distancing was an emotionally provoking and draining experience for Danish young adults. As highlighted by Bavel et al. [25] in their review of the social and psychological factors affecting COVID-19 pandemic behavior, “distancing clashes with the deep-seated human instinct to connect with others. Social connections help people regulate emotions, cope with stress and remain resilient during difficult times”. In this study, when faced with restrictions on social interaction, students felt physically and emotionally detached, which made them less able to cope with changes. One FHS in this study took an active approach to teach about alternatives to close physical interaction without losing social connections. Through discussions, they together cultivated an awareness among staff and students about the cultural diversity in social behavior including differences in ‘body cultures’ and ‘party and alcohol cultures’. This is an example of how FHSs and other educational institutions can facilitate a reflective process and experiment with learning and practicing new and alternative ways of showing affection, appreciation, attention, and participation. As Bavel et al. [25] put it, they can highlight the fact that social connection is possible even when people are physically distanced.


**Issue 4: How can behavioral sciences assist institutional settings in addressing future risk communication?**


We will here underline three aspects of how behavioral sciences, including behavior change communication, may inspire future risk communication at boarding-type schools, taking some key lessons learned from the Danish FHSs.

*Positive and pro-democratic approaches:* As highlighted by Bonell et al. [26], taking an authoritarian or forceful approach to behavior change is not effective for a long-term acceptance of behaviors. As further stressed by Wistoft [7], the tendency to moralize over young people not complying with COVID-19 guidelines, that have been observed in many countries, risks creating even more hostility towards COVID-19 guidelines among young adults. Positive and pro-social messages that create a ‘we’ are therefore recommended [12,25,26]. The general collective narrative issued by the authorities of ‘showing a collective mind’ was not effective, while specific and co-created metaphors identified in the FHSs in this study are good examples of ‘narratives that unite’.

Furthermore, the task given by law to the FHSs is to ‘foster democratic education’ of young adults, and contemporary views on FHSs stress the need for fostering pedagogical, political and societal awareness, responsibility, and the participation of students [10]. Lessons learned from previous pandemics state that engaging young people with the most familiar and culturally appropriate language and interventions is of high relevance for preventing the spread of disease [27]. Thus, we think FHSs and other boarding schools are well situated to engage in a non-judgmental, pro-democratic, and creative approach to future crisis-communication—in which young adults are actively engaged in co-creating, adapting, and implementing guidelines and new social norms.

*Credibility of communication:* Behavioral science also recommends that rules are communicated through channels and sources that are trustworthy and credible for the target audience [25,26]. Unclear and conflicting messages will risk impacting the credibility of messages negatively [12]. In this research we saw a tendency of young adults to be critical towards guidelines communicated by political and health authorities, since these authorities were not perceived as ‘knowledgeable’ about the ‘FHS spirit and life’ and context. We therefore recommend that educational institutions activate relevant associations and channels of communication that students find credible when communicating restrictions and guidelines to students, such as alumni groups, membership groups, local associations, etc. We also recommend that institutions attempt to coordinate key messages across institutions to prevent confusion and mistrust.

*Creative communication approaches:* Finally, institutional settings such as FHSs have a unique possibility to experiment with and develop effective and creative risk communication. Scandinavian FHSs are particularly well placed and skilled for this task, since they traditionally apply a rich diversity of creative educational approaches to adult learning [10]. However, in this study only two FHSs applied novel behavior change communication in the form of behavior design and nudging messages. We see a huge need and potential for more effective behavior change communication in educational authorities and institutions and recommend that they engage with creativity to design strategic and novel communication tools and methods for future risk communication.

## 5. Conclusions

In this study we applied the three concepts of sense of coherence, sense of community, and sense-making to understand how young Danish adults in an educational facility reacted to the restricted and uncertain reality during the COVID-19 pandemic. We found that COVID-19-related hygiene guidelines intuitively make sense, are socially acceptable for students, and are easy to implement for educational institutions. We therefore recommend an increased focus on improving hygiene in all types of boarding school institutions, also in the post-COVID-19 era, to prevent the spread of a wide range of infectious diseases and increase wellbeing. The study also demonstrates that educational institutions must acknowledge that epidemic guidelines may be perceived by young people as ‘non-sense’ partly due to feelings of loss of community, coherence, and freedom. Educational institutions such as FHSs should engage actively in normalizing processes to assist young people in re-building a new sense of coherence and community during and after a crisis. Danish FHSs draw on educational traditions that make them ideally placed for experimenting with creative and strategic risk communication to assist young adults in developing new coping strategies during crisis.

## Figures and Tables

**Table 1 ijerph-20-02557-t001:** Students’ answers to open survey question: ‘What could motivate you to follow the COVID-19 guidelines even better?’ (*n* = 357).

THEMES (Number of Students Who Mentioned the Theme in Open Survey Question; % of total Number of Students)	Number of Students (% of Total Number of Students Who Mentioned the Theme)
SOFTENING SOCIAL AND PHYSICAL DISTANCING GUIDELINES (*n* = 120; 33,6%)	
If the FHS could be a ‘closed bubble’ so we could avoid following all the restrictions regarding distancing and groupings	63 (52.5)
If the family groups were bigger and I was allowed to interact closely with more people	48 (40.0)
If we could be allowed to party and have social events again	9 (7.5)
SENSE-MAKING OF GUIDELINES (*n* = 90; 25.2% of total)	
If the restrictions were more meaningful/logical/clear to me	70 (77.8)
If rules were more practically adapted to fit our daily schedules	13 (14.4)
Clearer explanations of the restrictions and guidelines	3 (3.3)
More knowledge and information about the virus, infection, disease etc.	4 (4.4)
CREDIBILITY OF GUIDELINES (*n* = 47; 13.2% of total)	
If restrictions were purely based on health scientific evidence rather than being politically influenced	11(23.4)
If the rules were reviewed and adapted on a running basis/lifted when possible/ to fit the current COVID-19 situation in the country	19 (40.4)
If rules at the FHS mirrored ‘outside society’ more	8 (17.0)
If there was actual proof that restrictions are effective/if I can know for certain that rules will make a positive difference	9 (19.1)
HIGHER PERCEPTION OF THREAT (*n* = 32; 8.9% of total)	
If the danger/risk/threat from corona was higher/more present/if I thought it was more serious to my situation/my health	32 (100.0)
SOLIDARITY (*n* = 32; 8.9% of total)	
If everybody at the FHS would follow the guidelines rigorously/if we all took equal responsibility to follow them	31 (96.9)
If I know I take care of others in society (by following rules)	1 (3.1)
PUNISHMENT/CONSEQUENCES OF BREAKING RULES (*n* = 22; 6.2%)	
If there were more consequences/punishments if breaking the rules/if teachers would be more forceful and control that rules are followed	16 (72.7)
If I know that there is a danger/risk of me losing my place at the FHS/if the rules can ensure my stay at the FHS	6 (27.3)
OTHER MOTIVATIONS (*n* = 14; 3.9%)	
If we had better physical facilities at the FHS (which could ensure better distancing etc.)	4 (28.6)
If we could use ‘common sense’ instead of ‘policing’ of behaviors	3 (21.4)
A shift of mentality among students	1 (7.1)
That students are involved in creating rules	1 (7.1)
A hope that I can give hugs again soon	1 (7.1)
If we sat in smaller groups (less risk of spread of infection)	1 (7.1)
That rules at FHSs are not the same as at secondary boarding schools—we are grown-ups	1 (7.1)
If we all tested negative before arrival	1 (7.1)
If I could choose my own family group	1 (7.1)

**Table 2 ijerph-20-02557-t002:** Students’ sense-making of the guidelines.

Title of Theme	Aspects of ‘Sense Making’ within Theme	Quotations
Low credibility of guidelines	Guidelines are seen as ’nonsense’ due to a lack of credibility and mistrust in the issuing authorities; the authorities are perceived as ‘foreigners’ and ‘their’ guidelines are seen as incompatible with and disturbing to the ‘sense of FHS community’.	*“These guidelines are out of proportions! It seems as if those in the government have just decided; these FHSs, they are infectious. They just need to be closed down. They have ZERO sense of what’s going on here, they have no idea how a FHS functions. It’s so clear from those guidelines that they have no idea what they are talking about.”* (FGD, man, FHS3) *“I have to say, these guidelines really show signs of some old men or old men and women, who sit there and have some vague idea of what FHS is about and are forcing some rules down our throats. I would like it if some of us students could be involved.”* (FGD, woman, FHS4)
Unclarity and inconsistency of communication	Guidelines are perceived as ‘meaningless’ and incomprehensible due to a lack of clarity and consistency—and therefore ‘unmanageable’. But mainly, the new rules are challenging the old ‘common-sense thinking’ and force a new way of making sense.	Student 1: *“It’s quite hard to navigate in these rules. Because your ‘common sense‘ goes against it. There are so many things that doesn’t fit. For example, I have to clean a room that isn’t used at all.”* Student 2: *“And we used to have a sauna. Now it’s illegal to use it., But I still have to clean it! So I have to go down there and open it and clean it, even though no one uses it and no one has the key to it. This seems very pointless! (both laughing)”* (FGD, man and woman, FHS4) *”It’s so hard to follow the rules when you don’t really understand why they are here. And I felt like that from the very beginning. It would have been so much easier for me to comply with rules if I had thought to myself that there were really good reasons for them.”* (FGD, woman, FHS3).
Democracy versus individual freedom	Students feel the democratic dilemma during times of epidemics. The rules are challenging individual freedom at FHS, but also enforcing a new ‘sense of community’ building on trust and social responsibilities—values that are at the core of the ‘FHS community’.	*“I think the reason for the conflict is like FHSs, they don’t really do rules, like it’s about trust and personal responsibility […] And so I think it’s (making rules) just very against the whole philosophy of a FHS.”* (FGD, woman, FHS2). *“What they (the management at schools) are doing is that they’re handing the responsibility to us. So they’re trying to make us follow the guidelines and to be positive and very appreciative and trustful.”* (FGD, woman, FHS1) *“I also feel like letting us have the responsibility is… I think it captures like the essence of a folk high school, because it’s not about teaching you a certain class or something. Its teaching about life and taking responsibility over something like this, that no one ever experienced before. It is really something to learn from.”* (FGD, woman, FHS1)

**Table 3 ijerph-20-02557-t003:** Students’ perceptions of physical distancing.

Theme Title	Aspects of ‘Sense of Community’ and ‘Sense-Making’ within Theme	Quotations
**Physical distancing is against the ‘FHS spirit’/sense of community**	The ‘old sense of community’ feels lost and destroyed. Old values of community were defined by physical togetherness, physical interaction, physical presence. Distancing feels unnatural, strange, and uncomfortable.	*“Well, I think that physical distancing is SO difficult to comply with. Its really the opposite of what folk high schools are about. Being on a FHS is about presence and being physical all the time….Everything is really strange now […] Now we are divided into smaller family groups. And I really think it sucks!”* (young man, FGD, FHS3) *“I think it’s hard. A huge part of being on this school is about being together […] And that’s hard when you have to keep a distance. It feels very unnatural in a place like this.”* (young woman. FGD, FHS5) *“Its just not easy in a FHS environment to keep away from others. A lot of the sense of community is lost.”* (Survey answer)
**Physical distancing rules are nonsense**	Distancing rules feel meaningless, without sense, pointless, and ridiculous; the rules are therefore seen as impossible to follow.	Student 1: *“I think ‘pointless’ is a good word for it. Cause we also have classes together, and you live in the same small hallway, but when you eat you have to keep your distance.”* Student 2: *“There are so many weird rules, but I think no one gets all of them!”* (FGD, FHS4) *”I don’t think it makes any sense to keep distance to each other. Now we have been back at the school for 2 weeks, and no one is experiencing symptoms. So I think it’s ridiculous that we are not allowed to hug each other.”* (Survey answer)
**Emotional reactions to physical distancing**	Not being able to interact physically is destroying the sense of community through feelings of ‘being connected’ with friends, and being able to care for others naturally and without awkward distancing. This creates emotional distress including anger, sadness, loneliness, hopelessness, and anxiety.	*“I felt I had many friendships which were very you know—physical. When we had a conversation, we would stand closely or give a pad on the shoulder. I could feel that I didn’t really talk to that person properly, because I simply didn’t know how we should stand one meter apart and look at each other and like; ’How was your day?’ It became so awkward, and I could feel it in a real physical sense.”* (Young woman, FGD, FHS3) *“I experience that these rules are nothing but idiotic. That we only comply with them on paper, but in reality we are not able to walk with one meters distance when someone needs a hug. Its awkward to maintain these guidelines.”* (Survey answer) Anger: *“We had the first party after we came back. And we danced and we danced pretty close. And everybody was just like, oh I missed you so much! And one student took the microphone and said, guys you need to keep a distance because this is what the headmaster said that we are not supposed to do. And people became quite mad at him, and annoyed, and said not very nice things to him.”* (Woman, FGD, FHS 5) Sadness and loneliness: *“I think it’s really hard as it is now. Many of my friends are in another group than me. My roomie has also moved. So now I often feel lonely and sad. I can get angry because I am helpless. Authorities have decided to do these restrictions and I feel that they are far too pervasive, but I can’t do anything. The future is uncertain. And therefore I could use the free space of the FHS and to be close to my friends.” (survey answer)* Demotivation and anxiety: *“Nothing can motivate me to follow the rules. It doesn’t really feel like real folk high school anymore. The rules have given me anxiety.”* (Survey answer)

**Table 4 ijerph-20-02557-t004:** Students’ views on hygiene guidelines.

Theme	Quotations
**Hygiene makes sense and is the new normal**	*”It quite quickly became normal to clean, which I think is a very positive thing. Before corona-time we only cleaned the toilets like twice a week, so I really think it has benefitted us in many areas.”* (Young man, FGD, FHS4) Student 1: *”The most important thing is just to clean… and to wash hands every time you enter the dining hall or other places, where many people touch things. I have been thinking, this rule I just have to comply with.”* Student 2: *”Yes, that was just a condition when we came back to the school, and it’s just the way it is.”* (2 women, FGD, FHS4)
**Hygiene rules are easy to comply with**	*“I think that the hygiene part of rules, are the easiest to comply with… Its right in front of the dining hall, so you just wash your hand before you enter. And there is hand sanitizer when you enter the gym, and then you use the hand sanitizer. Its so easy to comply.”* (Young man, FGD, FHS3) Student 2: *“So it’s not really impacting your day [that you have to spend this time doing cleaning]. And how to, like, disinfect your hands—that takes 5 s and the washing stations are everywhere. It’s really easy. So it’s not a big deal.”* (FGD, FHS2)
**Learning about cleaning and gaining a cleaner living environment**	Student 1: *”I thought we should also talk about the good things. Now we have complained a lot. But in general, the cleanliness, that we could learn from, I think.”* Student 2: *“Yes, we have learned a lot from that. It’s so lovely, and actually, being forced to clean the toilets twice a day, then you actually really learn.”* Student 1: *“Yes, and now it’s shining!”* Student 2: *“And it’s an entirely different environment we live in. It’s so great.”* (FGD 2 young women, FHS4)
**International students’ views on hygiene**	*“During the quarantine here at the school and now, we got used to cleaning everything just after using it. And I hope that becomes common sense now. That is a way to respect the things. But people don’t clean too much here at this FHS. Actually! (laughing) So I was thinking about this. The gym was always a mess. But now it’s very clean. So I like it now. It is always like that in my country, so it was a culture shock for me to come here!”* (International student, FGD, FHS3)

## Data Availability

The datasets generated and analyzed during the current study are not publicly available due to its character (qualitative interviews, details about observations, and survey responses may be traceable to the person interviewed or the specific setting). Data from focus group discussions may be available in fully anonymized form from the corresponding author on reasonable request. Data from the survey is not publicly available since publication of this data is planned.

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
