# Peer review of "Making Sense or Non-Sense? Communicating COVID-19 Guidelines to Young Adults at Danish Folk High Schools"

_ijerph, 2023, doi:10.3390/ijerph20032557_

Round 1
Reviewer 1 Report
I found this article super interesting. However, here are some comments that should be plugged in in order to increase the viability of the results:
- The paper connect to behaviorial sciences but can we really draw from your materials any recommendations on communication in times of a pandemic. How do you make sense of the huger literature on the use of mobile phones or social networks ? I think about two references that might be of interest: A PNAS paper by Katherine Milkman & co-authors "A 680,000 mega study..." (published in 2021), and a PLOS ONE paper by Thomas Renault & co-authors (2021) on "Social distance beliefs and mobility" using Twitter indices. It also deals with partisanship which might be of importance in your context as you mention democracy.
- My second thought was about the important role of culture. I believe that works by Joe Heinrich from Harvard or Simon Porcher's work on culture and the quality of government (Public Administration Review, 2021) might be of interest for you as it shows how culture might be more or less enclined to deal with the COVID-19 situation. Communication might also completely differ depending on individualism / collectivism. As you are focused on a single country, I am wondering what is the generalizability of your results.
Author Response
Reviewer 1:
- I found this article super interesting. However, here are some comments that should be plugged in in order to increase the viability of the results: The paper connect to behaviorial sciences but can we really draw from your materials any recommendations on communication in times of a pandemic. How do you make sense of the huger literature on the use of mobile phones or social networks ? I think about two references that might be of interest: A PNAS paper by Katherine Milkman & co-authors "A 680,000 mega study..." (published in 2021), and a PLOS ONE paper by Thomas Renault & co-authors (2021) on "Social distance beliefs and mobility" using Twitter indices. It also deals with partisanship which might be of importance in your context as you mention democracy.
Response: Thank you – we are pleased that reviewer 1 finds the manuscript to be of relevance. We acknowledge that large scale digital studies in urban settings have been conducted on youth mobility and issues of physical distancing (for example the above mentioned paper by Milkman et al. and Renault et al) - factors that are also addressed in the present article. However, the Folk High School context, a boarding school type, that we focused on in the present paper, is a very specific physical and dense context that we do not find comparable with the above mentioned, large scale contexts. We have therefore chosen not to engage with this specific line of research.
- My second thought was about the important role of culture. I believe that works by Joe Heinrich from Harvard or Simon Porcher's work on culture and the quality of government (Public Administration Review, 2021) might be of interest for you as it shows how culture might be more or less enclined to deal with the COVID-19 situation. Communication might also completely differ depending on individualism / collectivism. As you are focused on a single country, I am wondering what is the generalizability of your results.
Response: We highly appreciate this observation, and agree that the concept of ‘culture’ is relevant to draw on when understanding how emergencies such as epidemic outbreaks are handled in different contexts. We have added this aspect to the discussion (page 13, issue 2) and also added some suggestions to which educational contexts, we see as comparable to Folk High Schools in Denmark – thus addressing the issue of generalizability. As we see it, any type of boarding school facilities in western individualistic cultural contexts may face similar paradoxes to governing behaviours during emergencies. This has also been specified in the text (page 13).
Reviewer 2 Report
1. In the section of the introduction, I suggest the author should focus on making sense or non-sense rather than the Folk High Schools. The author should provide more theoretical discourse about making sense or non-sense as the analytical tools.
2. In the section of the research aims, I suggest the author should provide more information about the communicative divide of COVID-19 among these students and how to reduce this information divide.
3. In the section of the method, I suggest the author should provide more theoretical kinds of literatures about the research tools, such as interviewing questions or questionnaires.
4. In Table 2, I suggest the author should provide more information about the theoretical discourse or explorative considerations of the themes.
5. In the section of the students’ views, I suggest the author should revise these statements or tables to more clean or precise analytical arguments.
6. In the section of the conclusion, I suggest the author should provide more information on theoretical and practical reflections on the analysis of making sense or non-sense of COVID-19 communications.
Author Response
Reviewer 2:
- In the section of the introduction, I suggest the author should focus on making sense or non-sense rather than the Folk High Schools. The author should provide more theoretical discourse about making sense or non-sense as the analytical tools.
Response: We thank reviewer 2 for this relevant comment and have added more depth to the theoretical section (section 1.4, page 3). We are drawing on the concepts of Sense of Coherence and Sense of Community, as originating from Antonovsky and later applied on cases of emergencies such as the COVID-19 pandemic.
- In the section of the research aims, I suggest the author should provide more information about the communicative divide of COVID-19 among these students and how to reduce this information divide.
Response: In the section of research aim we have specified that the study investigates how adults makes sense of infectious control measures and restrictions communicated by authorities and management within an educational setting. We have also specified that these insights can provide new perspectives on how to design future communication strategies in institutional settings, including how to make sense and address resistance, uncertainty and frustrations among youth.
- In the section of the method, I suggest the author should provide more theoretical kinds of literatures about the research tools, such as interviewing questions or questionnaires.
Response: A new section has been added on page 3 (line 126-132), specifying that we apply a pragmatic paradigm to the research study, including relevant references to this. In section 2.3 (page 4, line 184-) on data analysis, we have furthermore specified how this paradigmatic approach influenced the analysis of data.
- In Table 2, I suggest the author should provide more information about the theoretical discourse or explorative considerations of the themes.
Response: This recommendation has been followed for table 2 as well as table 3, by adding another column that expands on which theoretical aspects emerged from each theme.
- In the section of the students’ views, I suggest the author should revise these statements or tables to more clean or precise analytical arguments.
Response: This recommendation has been followed for all tables: Some quotes have been shortened, some have been omitted to make points clearer. The above mentioned addition of a ‘theory column’ to tables 2 and 3 is hopefully also adding to the clarity of the arguments.
- In the section of the conclusion, I suggest the author should provide more information on theoretical and practical reflections on the analysis of making sense or non-sense of COVID-19 communications.
Response: This recommendation has been followed; We have added more text in the conclusion to reflect on the role of ‘sense and non-sense’ in the analysis and how these concepts are useful in understanding youth’s reactions.
Reviewer 3 Report
Technically, this is a good article on the investigation of reactions of young adults 9 residing at Danish Folk High Schools (FHSs) towards COVID-19 guidelines and the communicative 10 styles used to enforce COVID-19 guidelines. The framework of "make sense" was relevant to the study. The findings (qualitative position) reveal that young adults reacted negatively when first faced with 13 the new reality of COVID-19 restrictions.
However, I feel that the study should be targeting reactions of Post-COVID-19 so that humans can apply the findings in terms of their survival because they have already survived the COVID-19 (locking the stable after the horses have been stolen). At the moment the study sounds as if it assumes that the COVID-19 will come back, therefore, humans should be educated to fight the COVID-19.
The Research Design and Methodology sections miss a discussion on a philosophical paradigm that support the Qualitative approach with relevant language. It sounds as if the study implicitly used a Pragmatic paradigm that combines qualitative and quantitative approaches. But there is no discussion on the paradigm.
Author Response
Reviewer 3:
- However, I feel that the study should be targeting reactions of Post-COVID-19 so that humans can apply the findings in terms of their survival because they have already survived the COVID-19 (locking the stable after the horses have been stolen). At the moment the study sounds as if it assumes that the COVID-19 will come back, therefore, humans should be educated to fight the COVID-19.
Response: Based on this comment, we have throughout the paper (introduction, analysis, discussion and conclusion) stressed that the findings of this research are not only relevant beyond the specific school context in Denmark, but also beyond the specific type of infectious emergency of COVID-19. In essence this paper investigates how Young adults’ in an educational context ‘makes sense of’ and reacts to communication about any type of emergency that represents new uncertain times for them.
- The Research Design and Methodology sections miss a discussion on a philosophical paradigm that support the Qualitative approach with relevant language. It sounds as if the study implicitly used a Pragmatic paradigm that combines qualitative and quantitative approaches. But there is no discussion on the paradigm.
Response: Thank you for pointing this out. This was also suggested by reviewer 2, point 3. See response under reviewer 2.
Round 2
Reviewer 1 Report
The authors implemented the suggestions.
Reviewer 2 Report
The author mostly incorporated my suggestions and revised the manuscript. The manuscript could be published in this form for the journal.